# Unsupervised Object Detection with Theoretical Guarantees

**Marian Longa**
Visual Geometry Group
University of Oxford
mlonga@robots.ox.ac.uk

**João F. Henriques**
Visual Geometry Group
University of Oxford
joao@robots.ox.ac.uk

## Abstract

Unsupervised object detection using deep neural networks is typically a difficult problem with few to no guarantees about the learned representation. In this work we present the first unsupervised object detection method that is theoretically guaranteed to recover the true object positions up to quantifiable small shifts. We develop an unsupervised object detection architecture and prove that the learned variables correspond to the true object positions up to small shifts related to the encoder and decoder receptive field sizes, the object sizes, and the widths of the Gaussians used in the rendering process. We perform detailed analysis of how the error depends on each of these variables and perform synthetic experiments validating our theoretical predictions up to a precision of individual pixels. We also perform experiments on CLEVR-based data and show that, unlike current SOTA object detection methods (SAM, CutLER), our method's prediction errors always lie within our theoretical bounds. We hope that this work helps open up an avenue of research into object detection methods with theoretical guarantees.

## 1 Introduction

Unsupervised object detection using deep neural networks is a long-standing area of research at the intersection of machine learning and computer vision. Its aim is to learn to detect objects from images without the use of training labels. Learning without supervision has multiple advantages, as obtaining labels for training data is often costly and time consuming, and in some cases may be impractical or unethical. For example, in medical imaging, unsupervised object detection can help save specialists' time by automatically flagging suspicious abnormalities [19], and in autonomous driving it may help automatically detect pedestrians on a collision course with the vehicle [3]. It is thus important to understand and develop better unsupervised object detection methods.

While successful, current object detection methods are often empirical and possess few to no guarantees about their learned representations. In this work we aim to address this gap by designing the first unsupervised object detection method that we prove is guaranteed to learn the true object positions up to small shifts, and performing a detailed analysis of how the maximum errors of the learned object positions depend on the encoder and decoder receptive field sizes, the object sizes, and the sizes of the Gaussians used for rendering. This is especially important in sensitive domains such as medicine, where incorrectly detecting an object could be costly. Our method guarantees to detect any object that moves in a video or that appears at different locations in images, as long as the objects are distinct and the images are reconstructed correctly.

We base our unsupervised object detection method on an autoencoder with a convolutional neural network (CNN) encoder and decoder, and modify it to make it exactly translationally equivariant (sec. 3). This allows us to interpret the latent variables as object positions and lets us train the network without supervision. We then use the equivariance property to formulate and prove a theorem that relates the maximum position error of the learned latent variables to the size of the encoder and

38th Conference on Neural Information Processing Systems (NeurIPS 2024).

decoder receptive fields, the size of the objects, and the width of the Gaussian used in the decoder (sec. 4). Next, we derive corollaries describing the exact form of the maximum position error as a function of these four variables. These corollaries can be used as guidelines when designing unsupervised object detection networks, as they describe the guarantees of the learned object positions that can be obtained under different settings. We then perform synthetic and CLEVR-based [12] experiments to validate our theory (sec. 5). Finally, we discuss the implications of our results for designing reliable object detection methods (sec. 6).

Concretely, the contributions of this paper are:

1. An unsupervised object detection method that is guaranteed to learn the true object positions up to small shifts.
2. A proof and detailed theoretical analysis of how the maximum position error of the method depends on the encoder and decoder receptive field sizes, object sizes, and widths of the Gaussians used in the rendering process.
3. Synthetic experiments, CLEVR-based experiments, and real video experiments validating our theoretical results up to precisions of individual pixels.

## 2   Related Work

**Object Detection.** Object detection is an area of research in computer vision and machine learning, dealing with the detection and location of objects in images. Popular supervised approaches to object detection include Segment Anything (SAM) [13], Mask R-CNN [7], U-Net [18] and others [4]. While successful, these methods typically require large amounts of annotated segmentation masks and bounding boxes, which may be costly or impossible to obtain in certain applications. Popular unsupervised and self-supervised object detection methods include CutLER [23], Slot Attention [17], MoNet [2] and others [5]. These methods aim to learn object-centric representations for object detection and segmentation without using training labels. Finally, unsupervised object localisation methods such as FOUND [21] and others [22] aim to localise objects in images, typically using vision transformer (ViT) self-supervised features. Compared to both current supervised and unsupervised object detection and localisation methods, our work is the only one that has provable theoretical guarantees of recovering the true object positions up to small shifts. It also requires no supervision.

**Identifiability in Representation Learning.** Identifiability in representation learning refers to the issue of being able to learn a latent representation that uniquely corresponds to the true underlying latent representation used in the data generation process. Some recent works aim to reduce the space of indeterminacies of the learned representations, and thus achieve identifiability, by incorporating various assumptions into their models. Xi et al. [24] categorise these assumptions for generative models into constraints on the distribution of the latent variables and constraints on the generator function. Some of their categories include non-Gaussianity of the latent distribution [20], dependence on an auxiliary variable [9, 10], use of multiple views [16], use of interventions [1, 15], use of mechanism sparsity [14], and restrictions on the Jacobian of the generator [6]. In contrast, in our work we achieve identifiability by making our network equivariant to translations, imposing an interpretable latent space structure, and requiring the data to obey our theorem's assumptions.

## 3   Method

In this section we describe the proposed method for unsupervised object detection with guarantees. On a high level, our architecture is based on an autoencoder that is fully equivariant to translations, which we achieve by making the encoder consist of a CNN followed by a soft argmax function to extract object positions, and making the decoder consist of a Gaussian rendering function followed by another CNN to reconstruct an image from the object positions (fig. 1). In the following sections we describe the different parts of the architecture in detail.

**Autoencoder with CNN Encoder and Decoder.** We start with an autoencoder, a standard unsupervised representation learning model, consisting of an encoder network $\psi$ that maps an image $x$ to a low-dimensional latent variable $z$, followed by a decoder network $\phi$ that maps this variable back to an image $\hat{x}$, with the objective of minimising the difference between $x$ and $\hat{x}$. Typically, the encoder and decoder networks are parametrised by multi-layer perceptrons (MLPs) or convolutional neural networks (CNNs) paired with fully-connected (FC) layers, however neither of these parametrisations

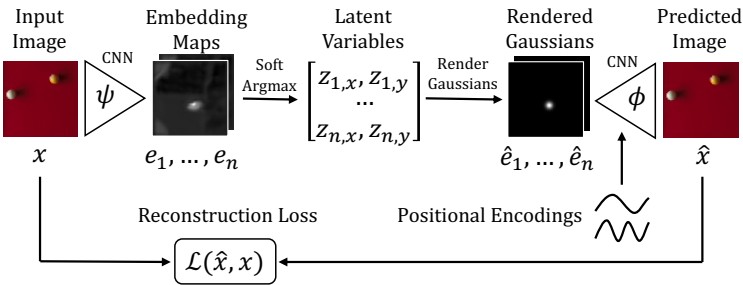

Figure 1: Network architecture. Encoder: (1) an image $x$ is passed through a CNN $\psi$ to obtain $n$ embedding maps $e_1, ..., e_n$, (2) a maximum of each map is found using softargmax to obtain latent variables $[z_{1,x}, z_{1,y}, ..., z_{n,x}, z_{n,y}]$. Decoder: (1) Gaussians $\hat{e}_1, ..., \hat{e}_n$ are rendered at the positions given by the latent variables, (2) the Gaussian maps are concatenated with positional encodings and passed through a CNN $\phi$ to obtain the predicted image $\hat{x}$. Finally, $x$ and $\hat{x}$ are used to compute reconstruction loss $\mathcal{L}(\hat{x}, x)$.

by default can guarantee that the learned latent variables will correspond to the true object positions (because of the universal approximation ability of MLPs and FC layers [8]). To obtain such guarantees, we would thus like to modify the autoencoder to make it exactly translationally equivariant, that is, a shift of an object in the input image $x$ should correspond to a proportional shift of the latent variable $z$, and a shift of the latent variable $z$ should correspond to a shift in the predicted image $\hat{x}$.

We start with an autoencoder where the encoder and decoder are both CNNs. CNNs consist of layers computing the convolution between a feature map $x$ and a filter $F$, defined in one dimension as

$$(x \star F)[i] = \sum_{j} x[j] F[j - i] \tag{1}$$

Intuitively, this corresponds to sliding the filter $F$ across the feature map $x$ and at each position of the filter $i$ computing the dot product between the feature map $x$ and the filter $F$. We can prove that convolutional layers are equivariant to translations, since

$$((\tau \circ x) \star F)[i] = \sum_{j} x[j - t] F[j - i] = \sum_{j} x[j] F[j - (i - t)] = \tau \circ (x \star F)[i] \tag{2}$$

where $\tau$ is the translation operator that translates a feature map by $t$ pixels, and we have used the substitution $j \to j + t$ at the second equality. Therefore, the encoder and decoder are both equivariant to translations, but this property only holds for translations of feature maps (i.e. spatial tensors).

**From Encoder Feature Maps to Latent Variables.** So far we have only worked with images and feature maps, but the latter do not directly express positions of any detected objects. It would be preferable to convert these feature maps into scalar variables that can be interpreted as object positions that are equivariant to image translations. To do this, we first define a translation $\tau$ of a (1D) feature map $x$ and a translation $\tau'$ of a scalar $z$ as

$$\tau(x)[i] = x[i - t], \quad \tau'(z) = z + t \tag{3}$$

where $i$ is the position in the feature map $x$, $\tau$ shifts an image by $t$ pixels, and $\tau'$ shifts a scalar by $t$ units. To relate translations in feature maps to translations in latent variables, we can use a function that computes a scalar property of a feature map $x$, such as $\mathrm{argmax}$, defined as $\mathrm{argmax}(x) = \{i : x[j] \leq x[i] \; \forall j\}$. Using these definitions we can now prove the equivariance of $\mathrm{argmax}$, i.e. that shifting the feature map $x$ by $\tau$ corresponds to shifting the latent variable $\mathrm{argmax}(x)$ by $\tau'$:

$$\mathrm{argmax}(\tau \circ x) = \{i : \tau \circ x[j] \leq \tau \circ x[i] \; \forall j\} = \{i : x[j - t] \leq x[i - t] \; \forall j\}$$
$$= \{i + t : x[j] \leq x[i] \; \forall j\} = \mathrm{argmax}(x) + t = \tau' \circ \mathrm{argmax}(x) \tag{4}$$

where at the first equality we use the definition of $\mathrm{argmax}$, at the second equality we use the definition of $\tau$ (eq. 3, left), at the third equality we use the substitution $i \to i + t$, at the fourth equality we use the definition of $\mathrm{argmax}$, and at the last equality we use the definition of $\tau'$ (eq. 3, right).

However, because the argmax operation is not differentiable, for neural network training we approximate it via a differentiable soft argmax function, defined in 2D as

$$\mathrm{softargmax}(x) = \left( \frac{1}{I} \sum_{i=0}^{I-1} \sum_{j=0}^{J-1} \left(i + \frac{1}{2}\right) \sigma_1 \left(\frac{x}{\Theta}\right)[i, j], \; \frac{1}{J} \sum_{i=0}^{I-1} \sum_{j=0}^{J-1} \left(j + \frac{1}{2}\right) \sigma_2 \left(\frac{x}{\Theta}\right)[i, j] \right) \tag{5}$$

where $\sigma$ is the softmax function defined in one dimension as $\sigma(x)[i] = \exp(x[i])/\sum_j \exp(x[j])$, $\sigma_1(x)$ and $\sigma_2(x)$ is the softmax function evaluated along the first and second dimensions of $x$, $\Theta$ is a temperature hyperparameter, $[i, j]$ is the image index, $I$ is the image width, $J$ is the image height, and the term $1/2$ ensures that the densities correspond to pixel centres. As the temperature $\Theta$ in eq. 5 approaches zero, $\mathrm{softargmax}$ reduces to the classical $\mathrm{argmax}$ function.

**From Latent Variables to Decoder Feature Maps.** Similar to mapping from encoder feature maps to latent variables, we would now like to relate shifts in latent variables $z$ to shifts of decoder feature maps $x$. To do this, we can invert the action of the $\mathrm{argmax}$ operation. Because $\mathrm{argmax}$ is a many-to-one function, finding an exact inverse is not possible, but we can obtain a pseudo-inverse using the Dirac delta function defined as $\mathrm{delta}(z)[i] = \delta(i - z)$. We can show that $\mathrm{delta}$ is a pseudo-inverse of $\mathrm{argmax}$ because $\mathrm{argmax} \circ \mathrm{delta} \circ z = i : \delta(j - z) \leq \delta(i - z) \, ; \forall j = z$. Now, similar to the $\mathrm{argmax}$ function, we can prove that the $\mathrm{delta}$ function is equivariant to the latent variable shift $\tau'$ on the input and the feature map shift $\tau$ on the output, i.e.

$$\mathrm{delta}(\tau' \circ z)[i] = \delta(i - \tau' \circ z) = \delta(i - z - t) = \mathrm{delta}(z)[i - t] = \tau \circ \mathrm{delta}(z)[i] \qquad (6)$$

where at the first equality we have used the definition of $\mathrm{delta}$, at the second equality we have used the definition of $\tau'$ (eq. 3, right), at the third equality we have used the definition of $\mathrm{delta}$, and at the last equality we have used the definition of $\tau$ (eq. 3, left).

Again, because the $\mathrm{delta}$ function is not differentiable, we can approximate it using a differentiable $\mathrm{render}$ function as

$$\mathrm{render}(z)[i] = \mathcal{N}(i - z, \sigma^2) \qquad (7)$$

where $\mathcal{N}(i - z, \sigma^2)$ is a Gaussian evaluated at $i - z$ with variance given by the hyperparameter $\sigma^2$. As the variance $\sigma^2$ in eq. 7 approaches zero, the $\mathrm{render}$ function reduces to the hard $\mathrm{delta}$ function.

Additionally, because the decoder is translationally equivariant, we also condition it on positional encodings of the size of the images to provide it with sufficient information to reconstruct different parts of the background, assuming the background is static. Alternatively, if background is varying, the decoder can be conditioned on a randomly-sampled nearby video frame instead, which will provide information about the background but not the objects' positions (following Jakab et al. [11]).

We also note that since the latent variables $z$ are ordered, this allows the encoder and decoder to learn to associate each variable with the semantics of each object and achieve successful reconstruction.

We thus now have all the elements we need to create an equivariant architecture where the encoder and decoder are defined, respectively, by

$$z = \mathrm{softargmax} \circ \psi \circ x, \quad \hat{x}_t = \phi \circ \mathrm{render} \circ z_t. \qquad (8)$$

This is shown in fig. 1. Having designed an exactly translationally equivariant architecture now allows us to obtain theoretical guarantees about the learned latent variables, which we discuss next. [1]

# 4 Theoretical Results

In this section we present our main theorem stating the maximum bound on the position errors of the latent variables learned with our method in terms of the encoder and decoder receptive field sizes, the object size, and the Gaussian standard deviation (thm. 4.1). We continue by deriving specialised corollaries relating the maximum position error to the encoder receptive field size (cor. 4.2), decoder receptive field size (cor. 4.3), object size (cor. 4.4), and Gaussian standard deviation (cor. 4.5).

**Theorem 4.1. Error Bound.** *Consider a set of images $x \sim X$ with objects of size $s_o$, CNN encoder $\psi$ with receptive field size $s_\psi$, CNN decoder $\phi$ with receptive field size $s_\phi$, soft argmax function $\mathrm{softargmax}$, rendering function $\mathrm{render}$ with Gaussian standard deviation $\sigma_G$ and $\Delta_G \sim \mathcal{N}(0, \sigma_G^2)$, and latent variables $z$, composed as $z = \mathrm{softargmax} \circ \psi \circ x$ and $\hat{x} = \phi \circ \mathrm{render} \circ z$ (fig. 1). Assuming (1) the objects are reconstructed at the same positions as in the original images, (2) each object appears in at least two different positions in the dataset, and (3) there are no two identical objects in any image, then the learned latent variables $z$ correspond to the true object positions up to object permutations and maximum position errors $\Delta$ of*

$$\Delta = \min\left(\frac{s_\psi}{2} + \frac{s_o}{2} - 1, \frac{s_\phi}{2} - \frac{s_o}{2} + \Delta_G\right). \qquad (9)$$

---

[1] We note that a similar architecture was proposed by Jakab et al. [11], with empirical success in keypoint detection. However, we derive our architecture by enforcing strict translation equivariance properties, which makes our theoretical results possible.

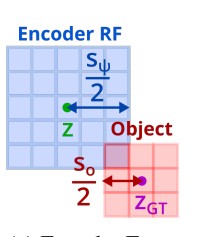 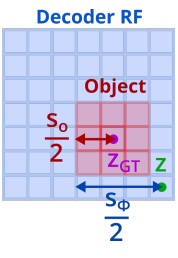

(a) Encoder Error          (b) Decoder Error

Figure 2: Position errors. (a) Maximum position error due to encoder, given by $s_\psi/2 + s_o/2 - 1$. The maximum error occurs when the encoder and the object are as far away from each other as possible while still overlapping by one pixel. (b) Maximum position error due to decoder, given by $s_\phi/2 - s_o/2 + \Delta_G$. The maximum error occurs when some part of the Gaussian at position $z + \Delta_G$ is within the decoder receptive field (RF) but is as far away from the rendered object as possible.

For proof see appendix A. Intuitively, the assumptions ensure that each latent variable corresponds to the position of each object in the image. The error in the learned object positions then arises from both the encoding and decoding process. In the encoding process, the maximum error occurs when the encoder and the object are as far away from each other as possible while still overlapping, i.e. $s_\psi/2 + s_o/2 - 1$ (fig. 2a). Conversely, in the decoding process, the maximum error occurs when the rendered object and the latent variable are as far away from each other as possible while both still being inside the decoder receptive field, i.e. $s_\phi - s_o/2$ (fig. 2b). Additionally, there is an extra error of $\Delta_G$ as the latent variable is rendered by a Gaussian and the decoder can capture any part of this Gaussian. Finally, because we assume each object is reconstructed at the same position as in the original image, the errors from the encoder and decoder must cancel each other out. Therefore, the overall maximum position error is given by the lower of the two expressions for the encoder and the decoder, leading to eq. 9. Next, we present corollaries relating this error bound to different factors.

**Corollary 4.2. Error Bound vs. Encoder RF Size.** *The maximum position error as a function of the encoder receptive field (RF) size $s_\psi$ for a given $s_\phi$, $s_o$, $\sigma_G$, is*

$$\Delta(s_\psi) = \begin{cases} \frac{s_\psi}{2} + \frac{s_o}{2} - 1 & \text{if } 1 \leq s_\psi \leq s_\phi - 2s_o + 2, \\ \frac{s_\phi}{2} - \frac{s_o}{2} + \Delta_G & \text{if } s_\psi > s_\phi - 2s_o + 2. \end{cases}$$

For an illustration see fig. 3a. There are two regions of the curve (separated by a dashed line). In the left-most region, $s_\psi < s_\phi - 2s_o + 2$, the error is dominated by the encoder error, and in the right-most region, $s_\psi \geq s_\phi - 2s_o + 2$, the error is dominated by the decoder error. Initially, for $s_\psi = 1$, the error is given by $s_o/2 - 1/2$, because the $1 \times 1$ px encoder can match any pixel that is part of the object and so can be at most half of the object size away from the true object position that is at the centre of the object. As the encoder RF size increases up to $s_\phi - 2s_o + 2$, the position error increases linearly with it as $s_\psi/2 + s_o/2 - 1$, because now any part of the encoder RF can match any part of the object (fig. 2a). This bound is deterministic due to the deterministic encoding process.

At $s_\psi = s_\phi - 2s_o + 2$ (vertical dashed line in fig. 3a), the maximum errors from encoder and decoder both become equal to $s_\phi/2 - s_o/2$. For $s_\psi > s_\phi - 2s_o + 2$, the position error is dominated by the error from the decoder which is constant at $s_\phi/2 - s_o/2 + \Delta_G$ with $\Delta_G \sim \mathcal{N}(0, \sigma_G^2)$, and so even though the encoder RF size is increasing, this has no effect as the limiting factor is now the decoder. Due to the Gaussian rendering step in the decoding process, this bound is now probabilistic, and is distributed normally with variance $\sigma_G^2$. The results of corollary 4.2 can be extended to multiple objects with different sizes (see appendix B, cor. B.1)

**Corollary 4.3. Error Bound vs. Decoder RF Size.** *The maximum position error as a function of the decoder receptive field (RF) size $s_\phi$ for a given $s_\psi$, $s_o$, $\sigma_G$, is*

$$\Delta(s_\phi) = \begin{cases} \frac{s_\phi}{2} - \frac{s_o}{2} + \Delta_G & \text{if } s_o \leq s_\phi < s_\psi + 2s_o - 2, \\ \frac{s_\psi}{2} + \frac{s_o}{2} - 1 & \text{if } s_\phi \geq s_\psi + 2s_o - 2. \end{cases}$$

For an illustration see fig. 3b. Similar to corollary 4.2, there are two regions of the curve, one for $s_\phi < s_\psi + 2s_o - 2$ (left), where the error is dominated by the decoder error, and another for $s_\phi \geq s_\psi + 2s_o - 2$ (right), where the error is dominated by the encoder error. Note that this is opposite to cor. 4.2. Initially, for $s_\phi = s_o$, the decoder receptive field has the same size as the object,

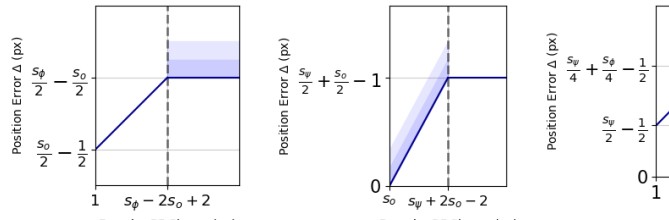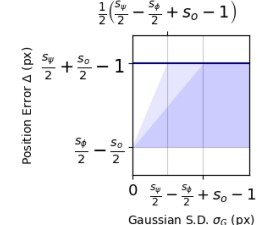

(a) Error vs. Encoder RF. (b) Error vs. Decoder RF. (c) Error vs. Object Size. (d) Error vs. Gaussian S.D.

Figure 3: Theoretical bounds for the maximum position error as a function of the encoder receptive field size $s_\psi$, decoder receptive field size $s_\phi$, object size $s_o$, and Gaussian standard deviation $\sigma_G$, as the remaining factors are fixed. Each bound consists of a region due to the encoder error (solid line) and the decoder error (probabilistic bound). Standard deviations are represented by shades of blue.

and so to achieve perfect reconstruction it needs to be at the same position as the object, resulting in 0 position error plus any error $\Delta_G$ caused by the non-zero width of the Gaussian. As the decoder RF size increases up to $s_\psi + 2s_o - 2$, the position error increases linearly with it as $s_\phi/2 - s_o/2 + \Delta_G$, because now the object can be at an increasing number of positions within the decoder and still achieve perfect reconstructions (fig. 2b). At $s_\phi = s_\phi + 2s_o - 2$, the maximum errors from encoder and decoder both become equal to $s_\psi/2 + s_o/2 - 1$. For $s_\phi > s_\psi + 2s_o - 2$, the position error is dominated by the error from the encoder which is constant at $s_\psi/2 + s_o/2 - 1$, and so even though the decoder RF size is increasing, this has no effect as the limiting factor is now the encoder. Similar to corollary 4.2, the results of corollary 4.3 can be extended to objects with multiple different sizes (see appendix B, cor. B.2).

**Corollary 4.4. Error Bound vs. Object Size.** *The maximum position error as a function of the object size $s_o$ for a given $s_\psi$, $s_\phi$, $\sigma_G$, is*

$$\Delta(s_o) = \begin{cases} \frac{s_\psi}{2} + \frac{s_o}{2} - 1 & \text{if } 1 \le s_o \le \frac{s_\phi}{2} - \frac{s_\psi}{2} + 1, \\ \frac{s_\phi}{2} - \frac{s_o}{2} + \Delta_G & \text{if } \frac{s_\phi}{2} - \frac{s_\psi}{2} + 1 < s_o \le s_\phi. \end{cases}$$

For an illustration see fig. 3c. Again, there are two regions of the curve, one for $s_o < s_\phi/2 - s_\psi/2 + 1$ (left), where the error is dominated by the encoder error, and one for $s_o \ge s_\phi/2 - s_\psi/2 + 1$ (right), where the error is dominated by the decoder error. Initially, for $s_o = 1$, the error is given by $s_\psi/2 - 1/2$, because any pixel of the encoder receptive field can match the $1 \times 1$ px object and so the error can be at most half of the encoder receptive field size. As the object size increases up to $s_\phi/2 - s_\psi/2 + 1$, the position error increases linearly with it as $s_\psi/2 + s_o/2 - 1$, because now any part of the encoder RF can match any part of the object (fig. 2a). At $s_o = s_\phi/2 - s_\psi/2 + 1$, the maximum errors from encoder and decoder both become equal to $s_\psi/4 + s_\phi/4 - 1/2$. For $s_o > s_\phi/2 - s_\psi/2 + 1$, the position error is dominated by the error from the decoder and decreases linearly as $s_\phi/2 - s_o/2 + \Delta_G$, because now there is a decreasing number of positions where the object can still fit inside the decoder receptive field (fig. 2b). At $s_o = s_\phi$, the object reaches the same size as the decoder, and thus the position error decreases to 0 with an additional error due to the width of the Gaussian, $\Delta_G$. Interestingly, the triangular shape of the error curve means that small and large objects will both incur small position errors, while medium sized objects will incur higher errors.

**Corollary 4.5. Error Bound vs. Gaussian Size.** *The maximum position error as a function of the Gaussian standard deviation $\sigma_G$ for a given $s_\psi$, $s_\phi$, $s_o$, is*

$$\Delta(\sigma_G) = \begin{cases} \frac{s_\phi}{2} - \frac{s_o}{2} + \Delta_G & \text{if } \sigma_G < \frac{s_\psi}{2} - \frac{s_\phi}{2} + s_o - 1, \\ \frac{s_\psi}{2} + \frac{s_o}{2} - 1 & \text{if } \sigma_G \ge \frac{s_\psi}{2} - \frac{s_\phi}{2} + s_o - 1. \end{cases}$$

For an illustration see fig. 3d. Firstly, there is an overall maximum bound for the position error due to the encoder, given by the constant $s_\psi/2 + s_o/2 - 1$, which is independent of the Gaussian standard deviation. Then, initially for $\sigma_G = 0$, the rendered Gaussian is effectively a delta function, and so the position error is dominated by the decoder error given by $s_\phi/2 - s_o/2$, which describes the maximum distance between the object and the delta function with both of them fitting inside the decoder receptive field (fig. 3b). As the Gaussian standard deviation increases, the position error increases linearly as $s_\phi/2 - s_o/2 + \Delta_G$ with $\Delta_G \sim \mathcal{N}(0, \sigma_G^2)$. Then, depending on what part of the Gaussian the decoder is convolved with, there are different bounds for the maximum

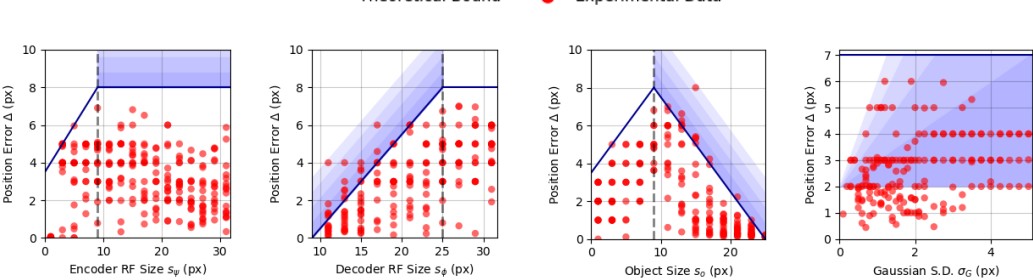

(a) Error vs. Encoder RF.    (b) Error vs. Decoder RF.    (c) Error vs. Object Size.    (d) Error vs. Gaussian S.D.

Figure 4: Synthetic experiment results showing position error as a function of the encoder receptive field size $s_\psi$, decoder receptive field size $s_\phi$, object size $s_o$, and Gaussian standard deviation $\sigma_G$, as the remaining factors are fixed to $s_\psi = 9, s_\phi = 25, s_o = 9, \sigma_G = 0.8$ (in a,b,c) or to $s_\psi = 9, s_\phi = 11, s_o = 7$ (in d). Theoretical bounds are denoted by a blue line (with 4 shaded regions denoting 1 to 4 standard deviations of the probabilistic bound) and experimental results by red dots.

position error. If the decoder is convolved with a part of the Gaussian that is within $n$ standard deviations of its centre, the maximum position error increases linearly as $s_\phi/2 - s_o/2 + n\sigma_G$ up to $\sigma_G = (s_\psi/2 - s_\phi/2 + s_o - 1)/n$, after which point the position error becomes dominated by the encoder value of $s_\psi/2 + s_o/2 - 1$. In fig. 3d, the maximum position error bound when the decoder is convolved with a part of the Gaussian within its first and second standard deviations, is denoted by darker and lighter shades of blue, respectively.

## 5   Experimental Results

In this section we validate our theoretical results on synthetic experiments (sec. 5.1) and CLEVR data (sec. 5.2). Additional real video experiments are in app. E. We first validate corollaries 4.2-4.4 via synthetic experiments in sec. 5.1, demonstrating very high agreement up to sizes of individual pixels. We then apply our method to CLEVR-based [12] data containing multiple objects of different sizes in varying scenes (sec. 5.2) and show that compared to current SOTA object detection methods (SAM [13], CutLER [23]), only our method predicts positions within theoretical bounds.

### 5.1   Synthetic Experiments

In this section we validate corollaries 4.2-4.5 via synthetic experiments. Our dataset consists of a small white square on a black background. In each experiment we fix all but one of the encoder RF size $s_\psi$, decoder RF size $s_\phi$, object size $s_o$, and Gaussian standard deviation $\sigma_G$, and vary the remaining variable. We perform each experiment 20 times, corresponding to 20 random initializations of the trained parameters, and record the position error $\Delta$ as the difference between the predicted object position $z$ and the ground truth object position $z_{GT}$. For more details see appendix C.1.

**Position Error vs. Encoder RF Size.** In this experiment we aim to empirically validate corollary 4.2, by measuring the experimental position errors as a function of the encoder receptive field size. We vary the encoder RF sizes $s_\psi \in \{1, 3, \ldots, 31\}$ and fix $s_\phi = 25, s_o = 9, \sigma_G = .8$ and record position errors $\Delta$. We visualise the data points (red) and the theoretical bounds (blue) in fig. 4a. We can observe that all the data points lie at or below the theoretical boundary, which validates corollary 4.2. In particular, we observe that the deterministic boundary in the region to the left of the dashed line (corresponding to the encoder bound) is well respected, with some of the trained networks achieving exactly the maximum error predicted by theory.

**Position Error vs. Decoder RF Size.** In this experiment we aim to validate corollary 4.3 by measuring the experimental position errors as a function of the decoder receptive field size. We vary the decoder RF sizes $s_\phi \in \{1, 3, \ldots, 31\}$ and fix $s_\phi = 9, s_o = 9, \sigma_G = .8$ and record position errors $\Delta$. We visualise the results in fig. 4b. The figure shows the theory to be a strong fit to the data, validating corollary 4.3. In particular, we note that the data points fit the Gaussian distribution in the decoder part of the curve (left of the dashed line) and are very close to (1 px below) the deterministic upper bound in the encoder part of the curve (right).

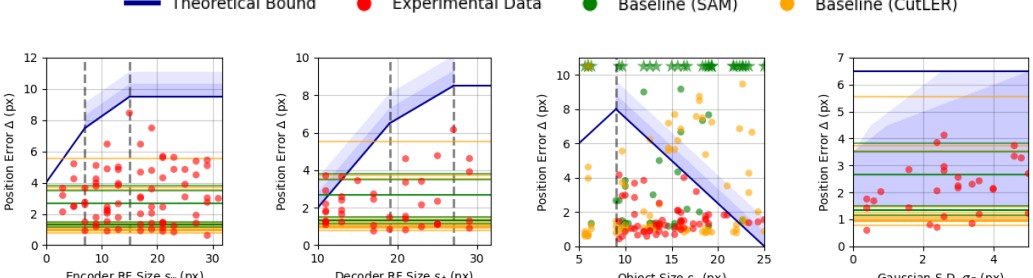

(a) Error vs. Encoder RF. (b) Error vs. Decoder RF. (c) Error vs. Object Size. (d) Error vs. Gaussian S.D.

Figure 5: CLEVR experiment results showing position error as a function of the encoder receptive field size $s_\psi$, decoder receptive field size $s_\phi$, and object size $s_o$, and Gaussian standard deviation $\sigma_G$, as the remaining factors are fixed to $s_\psi = 9, s_\phi = 25, s_o \in [6, 10], \sigma_G = 0.8$ for (a)-(c) and to $s_\psi = 5, s_\phi = 13, s_o \in [6, 10]$ for (d). Theoretical bounds are denoted by blue, experimental results in red, SAM baseline in green, and CutLER baseline in orange.

**Position Error vs. Object Size.** In this experiment we aim to validate corollary 4.4 by measuring the experimental position errors as a function of the object size. We vary the object sizes $s_o \in \{1, 3, \ldots, 25\}$ and fix $s_\psi = 9, s_\phi = 25, \sigma_G = .8$ and record position errors $\Delta$. We visualise the results in fig. 4c. As all the data points lie at or below the theoretical boundary, this validates corollary 4.4. We note that the empirical distribution of errors follows very closely the shape of the theoretical bound, very strictly on the left side of the dashed line (encoder bound) and according to the distribution predicted on the right side (decoder bound).

**Position Error vs. Gaussian Size.** In this experiment we aim to validate corollary 4.5 by measuring the experimental position errors as a function of the Gaussian standard deviation. We vary the Gaussian standard deviations $\sigma_G \in \{0.1, 0.2, \ldots, 2.1, 2.25, 2.5, ..., 5\}$, fixing $s_\psi = 9, s_\phi = 11, s_o = 7$ and record position errors $\Delta$. We visualise the data points (red) and the theoretical bounds (blue) in fig. 4d. As all the data points lie at or below the theoretical boundary, this validates corollary 4.5. In particular, we note that all the data points lie below the encoder bound (solid blue line), and all the data points lie within the bound denoted by four standard deviations away from the Gaussian. This means that in practice, the decoder can be convolved with any part of the Gaussian that lies within 4 standard deviations (corresponding to 3.2 px) from its centre. We also note that as the Gaussian standard deviation increases, the position error increases as expected, denoted by the positive slope of the data points between the third and fourth standard deviations (lightest shade of blue).

## 5.2 CLEVR Experiments

In this section we validate our theory on CLEVR-based [12] image data of 3D scenes. Our dataset consists of 3 spheres of different colours at random positions, with a range of sizes due to perspective distortion. We train and evaluate each experiment similarly to those in sec. 5.1, recording position errors for the learned objects, and compare our results with SAM [13] and CutLER [23] baselines. We compute the theoretical bounds according to our theory in sec. 4 and app. B, and visualise the results in figs. 5a-5c. For details see app. C.2. For experiments with different shapes see app. D.

Once again, the experimental results demonstrate high agreement with our theory, now even for more complex images with multiple objects and a range of object sizes (fig. 5, red, blue). Furthermore, while the SAM and CutLER baselines generally achieve low position errors, this is not guaranteed, and in some cases their errors are much higher than our bound (fig. 5, green, orange). We report the proportion of position errors from fig. 5c that lie within 2 standard deviations of our theoretical bound in table 6b and fig. 6a, showing that compared to SOTA object detection methods, only for our method are the position errors always guaranteed to be within our theoretical bound.

## 6 Discussion

In light of our theoretical results, in this section we present some conclusions that can be drawn when designing new unsupervised object detection methods:

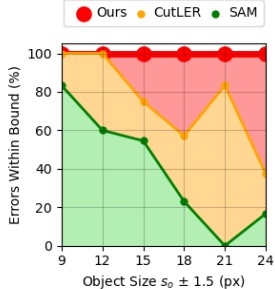

(a) Results visualisation.

| | Errors Within Bound (%) | | | | | | |
|---|---|---|---|---|---|---|---|
| | Object Size $\pm$ 1.5 (px) | | | | | | |
| Method | All | 9 | 12 | 15 | 18 | 21 | 24 |
| Ours | **100.0** | **100.0** | **100.0** | **100.0** | **100.0** | **100.0** | **100.0** |
| CutLER | 78.4 | 100.0 | 100.0 | 75.0 | 57.1 | 83.3 | 37.5 |
| SAM | 43.5 | 83.3 | 60.0 | 54.5 | 23.1 | 0.0 | 16.7 |

(b) Results table.

Figure 6: Proportion of position errors within 2 standard deviations of the theoretical bound (%), reported for different object sizes and methods. Results from table (b) are visualised in plot (a).

1. If the size of the objects that will be detected is known, to minimise the error on the learned object positions, one should aim to design the decoder receptive field size to be as small as possible while still encompassing the object. As the decoder RF grows beyond the object size, the error bound increases linearly with it up to a certain point (fig. 3b).

2. To minimise the error stemming from the encoder for a given object size, the encoder RF size should be kept as small as possible while still detecting the object (the RF size may be smaller than the object size), as again the error bound grows linearly with it up to a certain point (fig. 3a).

3. To minimise the error, the width of the rendering Gaussian should be kept as small as possible while still permitting gradient flow, as increasing it even slightly may result in a dramatic increase to the decoder term of the position error (fig. 3d). This is because, in practice, the decoder is able to detect parts of the Gaussian that are even 4 standard deviations away from its centre (fig. 4d).

4. In the case that one does not know *a priori* the exact size of the objects to be detected, one can still design a network that minimises the position errors for a given range of sizes. In that case, one should set up the decoder RF size to be as close as possible to the size of the largest object, and keep the encoder RF size as small as possible while still detecting all objects. The position errors for different object sizes will then be distributed according to the curve in fig. 3c, where the smallest and largest objects will achieve lowest errors and medium-size objects will achieve the greatest error, approximately given by a half of the average of the encoder and decoder RF sizes.

Finally, we discuss some limitations of our method. Firstly, the method can only detect dynamic objects, for example if they move in a video or if they appear at multiple locations in images. Secondly, in its current form the method learns representations that can not be used for videos with different backgrounds than the one used at training time; however, this can be overcome by conditioning the decoder on an unrelated video frame instead of the positional encodings, as in Jakab et al. [11]. Thirdly, the guarantees of our method are conditional on the images being successfully reconstructed, which depends on the network architecture and optimisation method.

# 7 Conclusion

We have presented the first unsupervised object detection method that is provably guaranteed to recover the true object positions up to small shifts. We proved that the object positions are learned up to a maximum error related to the encoder and decoder receptive field sizes, the object sizes, and bandwidth of the Gaussians used to render the objects. We derived expressions for how the position error depends on each of these factors and performed synthetic experiments that validated our theory up to sizes of individual pixels. We then performed experiments on CLEVR-based data, showing that unlike current SOTA methods, the position errors our method are always guaranteed to be within our theoretical bounds. We hope our work will provide a starting point for more research into object detection methods that possess theoretical guarantees, which are lacking in current practice.

**Acknowledgements.** The authors acknowledge the generous support of the Royal Academy of Engineering (RF\201819\18\163), the Royal Society (RG\R1\241385) and EPSRC (VisualAI, EP/T028572/1).

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

# A Proof of Theorem 4.1

***Theorem* Maximum Position Error**. Consider a set of images $x \sim X$ with objects of size $s_o$, CNN encoder $\psi$ with receptive field size $s_\psi$, CNN decoder $\phi$ with receptive field size $s_\phi$, soft argmax function $\mathrm{softargmax}$, rendering function $\mathrm{render}$ with Gaussian standard deviation $\sigma_G$ and $\Delta_G \sim \mathcal{N}(0, \sigma_G^2)$, and latent variables $z$, composed as $z = \mathrm{softargmax} \circ \psi \circ x$ and $\hat{x} = \phi \circ \mathrm{render} \circ z$ (fig. 1). Assuming (1) the objects are reconstructed at the same positions as in the original images, (2) each object appears in at least two different positions in the dataset, and (3) there are no two identical objects in any image, then the learned latent variables $z$ correspond to the true object positions up to object permutations and maximum position errors $\Delta$ of

$$\Delta = \min\left(\frac{s_\psi}{2} + \frac{s_o}{2} - 1, \frac{s_\phi}{2} - \frac{s_o}{2} + \Delta_G\right). \tag{10}$$

*Proof.* By assumption (1), the positions $(z_1, z_2)$ objects in the original image $x$ have to be the same as the positions $(\hat{z}_1, \hat{z}_2)$ of the objects in the reconstructed image $\hat{x}$. In practice, this occurs whenever the reconstruction loss is minimised.

By assumption (2) (each object appears at a minimum of 2 different positions), the latent variables used by the decoder have to contain some information about each object, and thus the encoder has to learn to match all the objects. This is because the decoder CNN $\phi$ takes as its input the rendered Gaussians $\hat{e} = \mathrm{render} \circ z$ concatenated with positional encodings or a randomly sampled nearby frame (fig. 1), and if some object in the dataset only appeared at a single position the model could achieve perfect reconstruction solely by using the positional encodings or the conditioned image without having to use the Gaussian maps $\hat{e}$. However, because the dataset contains each object at a minimum of 2 positions, relying purely on positional encodings or on the conditioned image is now not sufficient, as without any information about the object passed to the decoder, it would be impossible for it to predict where to render the object. More formally, because $\hat{x} = \phi \circ \mathrm{render} \circ \mathrm{softargmax} \circ \psi \circ x$, this means that for objects in $x$ to be reconstructed at the same positions as in $\hat{x}$ (assumption 1), the encoder $\psi$ needs to match some part of each object in $x$.

Because $z = \mathrm{softargmax} \circ \psi \circ x$ is equivariant to translations of $x$ (sec. 3), and because the encoder $\psi$ has to match some part of each object in $x$ (as shown previously), and also each image consists of distinct objects (assumption 3) on a known background, the image $x$ with an object at the position $(u_1, u_2)$ is encoded by $\mathrm{softargmax} \circ \psi$ to the latent variables

$$(z_1, z_2) = (u_1 + \Delta_{\psi 1}, u_2 + \Delta_{\psi 2}), \quad |\Delta_{\psi 1}|, |\Delta_{\psi 2}| \le \frac{s_\psi}{2} + \frac{s_o}{2} - 1 \tag{11}$$

where the shifts $\Delta_{\psi 1}$ and $\Delta_{\psi 2}$ arise because any part of the encoder filter (of size $s_\psi$) can match any part of the object (of size $s_o$). See fig. 2a for an illustration.

Next, because $\hat{x} = \phi \circ \mathrm{render} \circ z$ is equivariant to translations of $z$ (sec. 3), the latent variables $z = (u_1 + \Delta_{\psi 1}, u_2 + \Delta_{\psi 2})$ are mapped to a predicted image $\hat{x} = \phi \circ \mathrm{render} \circ z$ with an object at position

$$(\hat{z}_1, \hat{z}_2) = (u_1 + \Delta_{\psi 1} + \Delta_{\phi 1}, u_2 + \Delta_{\psi 2} + \Delta_{\phi 2}), \quad |\Delta_{\phi 1}|, |\Delta_{\phi 2}| \le \frac{s_\phi}{2} - \frac{s_o}{2} + \Delta_G \tag{12}$$

where the shifts $\Delta_{\phi 1}$ and $\Delta_{\phi 2}$ arise because any part of the decoder filter (of size $s_\phi$) can match any part of the the rendered Gaussian $\hat{e}_t = \mathrm{render} \circ z$, where $\Delta_G \sim \mathcal{N}(0, \sigma_G^2)$. See fig. 2b for illustration.

Finally, by assumption (1), the position of each object in the original image $x$ has to be equal to the position of the object in the reconstructed image, i.e.

$$(u_1, u_2) = (u_1 + \Delta_{\psi 1} + \Delta_{\phi 1}, u_2 + \Delta_{\psi 2} + \Delta_{\phi 2}) \tag{13}$$

This results in the conditions

$$\Delta_{\psi 1} + \Delta_{\phi 1} = 0, \quad \Delta_{\psi 2} + \Delta_{\phi 2} = 0 \tag{14}$$

and therefore

$$|\Delta_{\psi 1}| = |\Delta_{\phi 1}|, \quad |\Delta_{\psi 2}| = |\Delta_{\phi 2}| \tag{15}$$

In words, the shift in the latent variables acquired from the encoder $\Delta_\psi$ has to be balanced by an opposite shift of the same magnitude in the decoder $\Delta_\phi$ in order to reconstruct the object at the

same position. Because the shift due to the encoder is of maximum magnitude of $s_\psi/2 + s_o/2 - 1$ and the shift due to the decoder has a maximum magnitude of $s_\phi/2 - s_o/2 + \Delta_G$, this means that the maximum magnitude of the shift of the latent variables has to be the minimum of these two expressions, i.e. the learned latent variables $(z_1, z_2)$ correspond to the ground truth latent variables $(u_1, u_2)$ up to

$$(z_1, z_2) = (u_1 + \Delta_1, u_2 + \Delta_2), \quad |\Delta_1|, |\Delta_2| \leq \min\left(\frac{s_\psi}{2} + \frac{s_o}{2} - 1, \frac{s_\phi}{2} - \frac{s_o}{2} + \Delta_G\right). \quad (16)$$

Additionally, because the order in which the objects get mapped to each latent variable is arbitrary, there is an additional indeterminacy arising due to variable permutations. $\qquad\square$

# B    Theoretical Results for Multiple Object Sizes

The results of corollary 4.2 can be extended to objects with a range of different sizes. The bound can be obtained by taking the maximum over all the bounds for objects with sizes $s_o \in [s_o^{min}, s_o^{max}]$, leading to the following corollary.

**Corollary B.1. Error vs. Encoder RF Size for Multiple Object Sizes.** *The maximum position error as a function of the encoder receptive field (RF) size $s_\psi$ for a given $s_\phi$, $s_o \in [s_o^{min}, s_o^{max}]$, $\sigma_G$, is*

$$\Delta(s_\psi) = \begin{cases} \frac{s_\psi}{2} + \frac{s_o^{max}}{2} - 1 & \text{if } 1 \leq s_\psi \leq s_\phi - 2s_o^{max} + 2, \\ \frac{s_\psi}{4} + \frac{s_\phi}{4} - \frac{1}{2} + \Delta_G & \text{if } s_\phi - 2s_o^{max} + 2 < s_\psi \leq s_\phi - 2s_o^{min} + 2, \\ \frac{s_\phi}{2} - \frac{s_o^{min}}{2} + \Delta_G & \text{if } s_\psi > s_\phi - 2s_o^{min} + 2. \end{cases}$$

For an illustration see fig. 7a.

Similar to corollary 4.2, the results of corollary 4.3 can be extended to objects with a range of different sizes by taking the maximum over the bounds for objects with sizes $s_o \in [s_o^{min}, s_o^{max}]$, leading to the following corollary.

**Corollary B.2. Error vs. Decoder RF Size for Multiple Object Sizes.** *The maximum position error as a function of the decoder receptive field (RF) size $s_\phi$ for a given $s_\psi$, $s_o \in [s_o^{min}, s_o^{max}]$, $\sigma_G$, is*

$$\Delta(s_\phi) = \begin{cases} \frac{s_\phi}{2} - \frac{s_o^{min}}{2} + \Delta_G & \text{if } s_o^{min} \leq s_\phi \leq s_\psi + 2s_o^{min} - 2, \\ \frac{s_\psi}{4} + \frac{s_\phi}{4} - \frac{1}{2} + \Delta_G & \text{if } s_\psi + 2s_o^{min} - 2 < s_\phi \leq s_\psi + 2s_o^{max} - 2 \\ \frac{s_\psi}{2} + \frac{s_o^{max}}{2} - 1 & \text{if } s_\phi > s_\psi + 2s_o^{max} - 2. \end{cases}$$

For an illustration see fig. 7b.

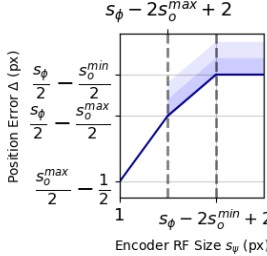

(a) Position error vs. encoder RF size for a range of object sizes.

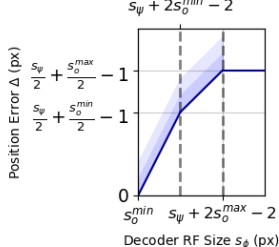

(b) Position error vs. decoder RF size for a range of object sizes.

Figure 7: Theoretical bounds for the maximum position error as a function of the encoder receptive field size $s_\psi$ and the decoder receptive field size $s_\phi$ for objects of sizes ranging from $s_o^{min}$ to $s_o^{max}$, as the remaining factors are kept constant. Each theoretical bound consists of two regions separated by a dashed line, one where the maximum error is due to the encoder (deterministic, represented by a solid line), and one where the maximum error is due to the decoder (probabilistic, represented by a solid line and shaded regions). Areas within one and two standard deviations of the mean are represented by a darker and lighter shades of blue respectively.

# C  Experiment Training Details

## C.1  Synthetic Experiments

**Dataset.** In all of the synthetic experiments, we use the following setup. The dataset consists of black images of size $s_{img} \times s_{img} = 80 \times 80$ px, each with a white square with dimensions $s_o \times s_o$ px, centered at positions $(x, y)$ where $x, y \in \{s_{pad} + s_o/2, s_{pad} + s_o/2 + 1, \ldots, s_{img} - s_{pad} - s_o/2\}$, where $s_{pad} \geq \max(s_\psi, s_\phi) - 1$ (to prevent unwanted edge effects). We divide the dataset into 4 quadrants and assign images from 3 quadrants to the training set and the remaining quadrant to the test set.

**Evaluation.** For each experiment, we fix all but one variable from the following set: encoder RF size $s_\psi$, decoder RF size $s_\phi$, object size $s_o$, and Gaussian standard deviation $\sigma_G$, and vary the remaining variable. For each value of the investigated variable we perform 20 experiments with different random seeds, noting down the learned position error $\Delta$ as the absolute difference between the learned position $z$ and the centre of the object $z_{GT}$ (maximum over horizontal and vertical differences, and over the test set). We discard a result if the reconstruction accuracy of the run is below $99.9\%$, to only consider runs where the object has been detected successfully (this is because the square is on the order of $7 \times 7$ px, thus only comprising $0.8\%$ of the image).

**Architecture.** We parametrise both the encoder $\psi$ and the decoder $\phi$ as 5-layer CNNs with Batch Normalisation and ReLU activations, 32 channels, and filter sizes in $\{1 \times 1, 3 \times 3, 5 \times 5, 7 \times 7\}$, such that their receptive field sizes equal $s_\psi$ and $s_\phi$ respectively. We train each network for 500 epochs using the Adam optimiser with learning rate $10^{-3}$ and batch size 128. We train each experiment on a single GPU for around 6 hours with <6GB memory on an internal cluster.

## C.2  CLEVR Experiments

**Dataset.** Our CLEVR experiments use data generated with the CLEVR [12] image generation script. Our training and test sets consist of 150 and 50 images respectively, containing red, green and blue metallic spheres on a random background, at random positions, and with a different range of sizes (fig. 8). For experiments measuring position error as a function of encoder and decoder RF sizes and Gaussian s.d., we use a dataset with object sizes between 6-10 px after perspective distortion (fig. 8a). For the experiment measuring position error as a function of object size, we use 5 datasets with object sizes 9-14 px, 11-17 px, 13-19 px, 15-24 px, 17-27 px after perspective distortion (figs. 8b, 8c).

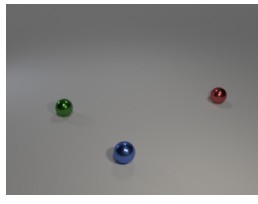 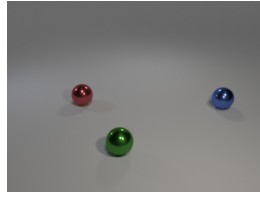 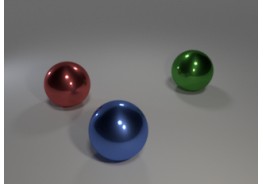

(a) Object sizes 6-10 px.    (b) Object sizes 9-14 px.    (c) Object sizes 17-27 px.

Figure 8: Samples from our CLEVR datasets, with object sizes (a) 6-10 px, (b) 9-14 px, (c) 17-27 px.

**Evaluation.** For each experiment, we fix all but one variable from the following set: encoder RF size $s_\psi$, decoder RF size $s_\phi$, object size $s_o$, and Gaussian standard deviation $\sigma_G$, and vary the remaining variable. For each value of the investigated variable we perform experiments with different random seeds, noting down the learned position errors $\Delta$ as the absolute difference between the learned position $z$ of each object and the centre of the object $z_{GT}$ (maximum over horizontal and vertical differences and over the test set, for each object). We only consider results for objects that have been learned successfully. For experiments measuring position error as a function of encoder and decoder RF sizes and Gaussian s.d., we consider an object to be learned if the position error is less than 35 px, and if only a single variable corresponds to the object, and if the position error is stable over consecutive training iterations. For the experiment measuring position error as a function of object size, we only consider runs where all 3 objects have been learned, i.e. where the reconstruction accuracy is higher than $98.0\%$ for the dataset with object sizes 6-10 px, $98.8\%$ for dataset with sizes

11-17 px, 98.0% for dataset with sizes 13-19 px, 97.5% for dataset with sizes 15-24 px, and 98.0% for dataset with sizes 17-27 px.

**Architecture.** We parametrise both the encoder $\psi$ and the decoder $\phi$ as 5-layer CNNs with Batch Normalisation and ReLU activations, 32 channels, and filter sizes in $\{1 \times 1, 3 \times 3, 5 \times 5, 7 \times 7\}$, such that their receptive field sizes equal $s_\psi$ and $s_\phi$ respectively. We train each network until convergence using the Adam optimiser with learning rate $10^{-2}$ and batch size 128 for the experiments measuring position error as a function of encoder and decoder RF size and Gaussian s.d., and with learning rate $10^{-3}$ and batch size 150 for the experiment measuring position error as a function of object size. We train each model on a single GPU for less than day with <8GB memory on an internal cluster.

**Baselines.** We also evaluate the results for two State-of-the-Art baselines, SAM [13] and CutLER [23]. First, for each of our 6 datasets (containing objects of sizes 6-10 px, ..., 17-27 px), we combine its training and test set to create 4 sets of 50 images each. We then apply SAM and CutLER to all images in each 50-image set, noting down the learned position errors $\Delta$ as the absolute difference between the predicted position $z$ of each object and the centre of the object $z_{GT}$ (maximum over horizontal and vertical differences and over the 50-image set, for each object). For SAM, we take the predicted object positions to be the centres of the bounding boxes corresponding to the second, third and fourth predicted masks (first mask corresponding to the background). For CutLER, we take the predicted object positions to be the centres of the predicted bounding boxes if the method predicts 3 bounding boxes (one for each object), otherwise we discard the prediction. For both methods, we discard any result where position error is greater than 35 px, to be consistent with the evaluation for our method. Finally, this results in 12 position error values (3 objects × 4 data splits), for each method and each of our 6 datasets.

## D CLEVR Experiments with Different Shapes

To demonstrate that our method applies to objects of any shape, in this section we include experiments on our CLEVR dataset with three distinct objects – a red metallic sphere, a blue rubber cylinder and a green rubber cube. The dataset has objects of size 9-19 px after perspective distortion, and a sample from the dataset is shown in fig. 9a. We perform training and evaluation in the same way as in app. C.2. We plot the position error as a function of the encoder and decoder receptive field sizes in figs. 9b and 9c, respectively. We can observe that all our experimental data (red) lies within the bounds predicted by our theory (blue), successfully validating our theory for objects with different shapes.

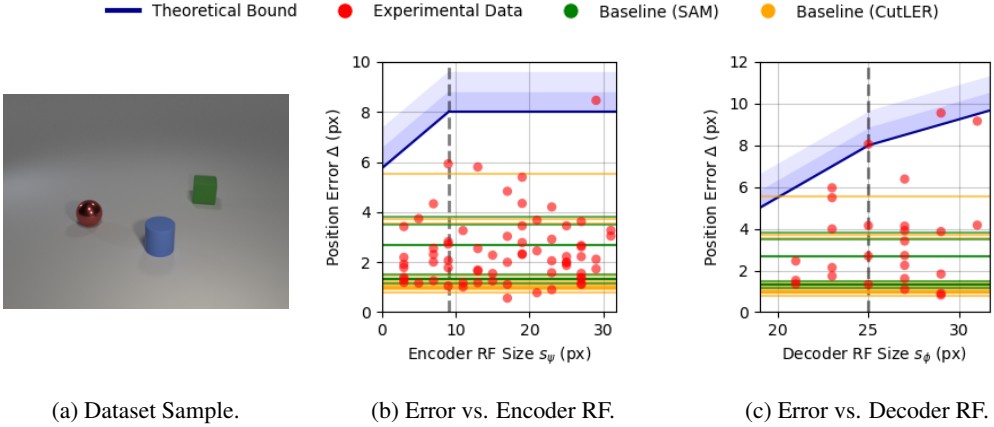

(a) Dataset Sample.      (b) Error vs. Encoder RF.      (c) Error vs. Decoder RF.

Figure 9: CLEVR experiment results using (a) a dataset with 3 objects of different shapes (a sphere, a cube, and a cylinder), showing position error as a function of (b) the encoder receptive field size $s_\psi$ and (c) decoder receptive field size $s_\phi$, as the remaining factors are fixed to $s_\psi = 9, s_\phi = 25$, object size $s_o \in [9, 19]$ and Gaussian s.d. $\sigma_G = 0.8$. Theoretical bounds are denoted by blue, experimental results in red, SAM baseline in green, and CutLER baseline in orange.

# E Experiments with Real Videos

In this section we present experimental results of applying our method to real YouTube videos, including overhead traffic footage and mini pool game footage.

## E.1 Traffic Data

In this experiment we aim to learn the position of a car from an overhead traffic video. The training and test sets for this experiment consist of 25 frames each, from a video of an overhead view of a car moving for a short distance in a single lane (fig. 10a). We train the architecture from fig. 1 on this training set and validate it on the test set. It achieves a mean squared error between the ground truth object positions and the learned object positions of $7.2 \cdot 10^{-5}$ (in units normalised by the image size), demonstrating that the object position has been learned successfully with a very low error. We then modify the learned position variable and decode it to generate videos of the car at novel positions (figs. 10b, 10c).

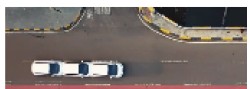
(a) Training data.

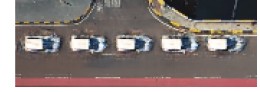
(b) Generated data (steady speed in a different lane).

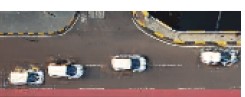
(c) Generated data (lane change and acceleration).

Figure 10: Road traffic video used for training, together with two videos generated after training by modifying and decoding the learned latent variables (video frames are superimposed). The car object is detected successfully and is used to generate realistic videos with objects at unseen positions.

## E.2 Mini Pool

In this experiment we aim to learn the positions of balls from a video of a game of mini pool. The training and test sets for this experiment consist of 15 and 11 frames respectively from a video of a game of mini pool, cropped to a portion where two balls are moving at the same time (fig. 11a). We train the architecture from fig. 1 on this training set and validate it on the test set, where it achieves a mean squared error between the ground truth object positions and the learned object positions of $8.2 \cdot 10^{-3}$ (in normalised units). This demonstrates that the object positions were learned successfully with a very low error. We then modify the learned position variables and decode them to generate videos of the balls at novel positions (figs. 11b, 11c).

In practice, it was important to set the encoder and decoder receptive field sizes to be greater than but close to the size of the objects, as for larger RF sizes the position error increased unnecessarily and the images were rendered further away from the position given by the latent variables. Also, for large receptive field sizes, the decoder filter contained too much background which caused low quality reconstructions when rendering the balls near the mini pool table edges.

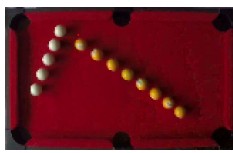
(a) Training data.

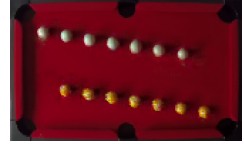
(b) Generated data (linear motion at unseen positions).

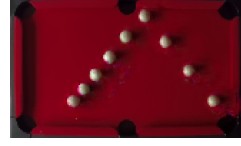
(c) Generated data (collision and slowing down).

Figure 11: Mini pool video used for training, together with two videos generated after training by modifying and decoding the learned latent variables (video frames are superimposed). Both ball objects are detected successfully and are used to generate realistic videos.

