# OpenReview forum: "Unsupervised Object Detection with Theoretical Guarantees"
_NeurIPS.cc/2024/Conference — NeurIPS 2024 poster_

### Official Review · Reviewer_ghv9 · 2024-06-30

**Soundness:** 3
**Presentation:** 4
**Contribution:** 3
**Rating:** 7
**Confidence:** 4

**Summary:**

This paper proposes an autoencoder based object detection model that makes predictions about object positions in an unsupervised manner. Imporantly, the authors can provide theoretical guarantees/ bounds about the degree of the model's detection error.

**Strengths:**

The paper is well written and it is easy to follow the authors motivation and structure of the work. I also find the idea very valuable to investigate the theoretical bounds of such models.

**Weaknesses:**

I have summarized my questions and issues and limitations that I see here:

In the context of the CLEVR experiments, I am wondering why the authors don’t evaluate concerning the Gaussian standard deviation as they did for the first dataset?

The authors claim the method requires dynamic objects, but they never mention this in main text. Can the authors provide more justification of this? I.e. why this is/ what part of the approach requires this?

I don’t understand how the decoder can learn to reconstruct complex shapes other than spheres (due to the Gaussian assumption). Also the authors mainly evalaute on data with objects that are spherical. Thus, is it possible to evaluate on other shape forms? If so what is the error here compared to spherical shapes? I do not mention this as a limitation, but it seems quite important to put the method into context. What would be potential ideas to mitigate handling more complex objects?

I do not have enough knowledge about the details of the CutLer and SAM models, but why should the theoretical bound of this work hold for these works as well (the authors compare these in Fig. 6)? Specifically, the authors state "only for our
method are the position errors always guaranteed to be within our theoretical bound." so my question is: why should the other methods lie within this theoretical guarantee?

I am a little confused by the related works section. The authors discuss object-centric representation methods whose goal, unlike that of their method, is to learn a full representation of an object. This includes much more information than just position. In other words, it seems the method of this paper focuses “only” on the learning the position of an object. While this does not diminish the significance of the work, I think the work could benefit from discussing more on this difference between these works, to make the comparisons more fair and also focus more on works that focus on unsupervisedly localising object in images (i.e., works that only focus on position and not on the other aspects of object reperesentations), e.g., [1,2]. So in the end I am also wondering if the authors should actually narrow down the title/contribution claims to "Unsupervised Object Localisation with Theoretical Guarantees"?

If the authors can remark on these issues above, I am happy to consider raising my score.

[1] Siméoni, Oriane, Chloé Sekkat, Gilles Puy, Antonín Vobecký, Éloi Zablocki, and Patrick Pérez. "Unsupervised object localization: Observing the background to discover objects." In Proceedings of the IEEE/CVF Conference on Computer Vision and Pattern Recognition, pp. 3176-3186. 2023.
[2] https://github.com/valeoai/Awesome-Unsupervised-Object-Localization

**Questions:**

see above

**Limitations:**

see above

---

> ### Author Rebuttal · Authors · 2024-08-05
>
> > In the context of the CLEVR experiments, I am wondering why the authors don’t evaluate concerning the Gaussian standard deviation as they did for the first dataset?
>
> We thank the reviewer for their suggestion and have performed this experiment – please see fig. 11 of the rebuttal PDF. As all our data points lie within our theoretical bounds, this successfully validates our theory.
>
> > The authors claim the method requires dynamic objects, but they never mention this in main text. Can the authors provide more justification of this? I.e. why this is/ what part of the approach requires this?
>
> We mention this in the Introduction: “Our method guarantees to detect any object that moves in a video or that appears at different locations in images”, and later in assumption 2 of Theorem 4.1: “each object appears in at least two different positions in the dataset”. We will reinforce it further throughout the paper that this is equivalent to requiring dynamic objects. This prevents the object being learned as part of the background – if it did not move, then the encoder would not need to communicate its position to the decoder to reconstruct every image in the dataset, and the object would not be detected. The proof in Appendix A has more details on how this assumption is used.
>
> > I don’t understand how the decoder can learn to reconstruct complex shapes other than spheres (due to the Gaussian assumption).
>
> The decoder takes as input binary maps containing Gaussians centered at the object positions given by the latent variables ($\hat{e}$ in fig. 1). The decoder, consisting of multiple convolution blocks, then iteratively convolves these Gaussians with its kernels (which are not Gaussian, but arbitrary/learned) until it reaches its receptive field size. For example, if the decoder contains 3 7x7 convolutions and the Gaussian bandwidths are narrow, this will result in the receptive field size of $1 + 3 \times (7-1) = 19$, allowing it to reconstruct any object of size up to 19x19 px, regardless of its shape (not just Gaussians).
>
> > Also the authors mainly evalaute on data with objects that are spherical. Thus, is it possible to evaluate on other shape forms? If so what is the error here compared to spherical shapes? I do not mention this as a limitation, but it seems quite important to put the method into context. What would be potential ideas to mitigate handling more complex objects?
>
> While in our CLEVR experiments in the paper we have used spherical objects, the method works on objects of any shape (see answer to previous question). We have performed CLEVR experiments using 3 different shapes – a sphere, a cube, and a cylinder – and show the results in fig. 12 of the rebuttal PDF. As all our data points lie within our theoretical bounds, this shows that our method is applicable to objects of any shape, not just spheres.
>
> > I do not have enough knowledge about the details of the CutLer and SAM models, but why should the theoretical bound of this work hold for these works as well (the authors compare these in Fig. 6)? Specifically, the authors state "only for our method are the position errors always guaranteed to be within our theoretical bound." so my question is: why should the other methods lie within this theoretical guarantee?
>
> The CutLER and SAM methods do not possess any theoretical guarantees on their object detections, and so there is no theoretical bound for their errors. We have only included this comparison to show that, even in the worst case (maximum over the position errors), the errors of our method are always bounded by our theoretical bound while the errors for CutLER and SAM are not, and are much higher than our bound in some settings. The experiment is meant to illustrate how worst-case unbounded errors can occur for such state-of-the-art methods, which can be a safety concern.
>
> > I think the work could benefit from discussing more on this difference between these works, to make the comparisons more fair and also focus more on works that focus on unsupervisedly localising object in images (i.e., works that only focus on position and not on the other aspects of object reperesentations), e.g., [1,2].
>
> We thank the reviewer for their suggestion and will discuss these works in the related work section of the paper. Briefly, works such as [1] and other works in [2] rely on vision transformer (ViT) self-supervised features for unsupevised object detection and segmentation, without possessing any theoretical guarantees for their detection errors. In contrast, our method uses a translationally equivariant CNN-based encoder and decoder with a structured bottleneck, which allows us to prove theoretical bounds for our detection errors.
>
> [1] Siméoni, Oriane, Chloé Sekkat, Gilles Puy, Antonín Vobecký, Éloi Zablocki, and Patrick Pérez. "Unsupervised object localization: Observing the background to discover objects." In Proceedings of the IEEE/CVF Conference on Computer Vision and Pattern Recognition, pp. 3176-3186. 2023.
>
> [2] https://github.com/valeoai/Awesome-Unsupervised-Object-Localization

---

> > ### Comment · Reviewer_ghv9 · 2024-08-10
> >
> > Thanks so much for the effort and clarification. I have raised my score accordingly.
> >
> > Concerning my point about other related work: I think the authors could think about differentiatting throughout the work on the difference on learning object representations and explicit object (position) detection. But, the response was fine. And I am convinced if the authors edit the main text as they had responded.

---

### Official Review · Reviewer_pKP5 · 2024-07-11

**Soundness:** 3
**Presentation:** 3
**Contribution:** 3
**Rating:** 6
**Confidence:** 4

**Summary:**

This paper explores Unsupervised Object Detection with Theoretical Guarantees. This method is a significant advancement in the field of object detection as it provides theoretical guarantees on the accuracy of the detected object positions. By introducing a new approach that ensures reliable object localization, the research contributes to enhancing the robustness and accuracy of unsupervised object detection systems.

**Strengths:**

The method provides theoretical guarantees on recovering true object positions up to small shifts, which is a significant strength compared to traditional empirical approaches in object detection. The ability to interpret the latent variables as object positions enhances the interpretability of the model and facilitates understanding of the learned representations. The use of an autoencoder with a convolutional neural network (CNN) encoder and decoder, modified to be translationally equivariant, offers a unique and innovative approach to unsupervised object detection.

**Weaknesses:**

This work explores the unsupervised object detection, and theoretical analysis. However, the dataset for the experiment is not common, and few comparative experiments with common SOTA object detection model. Besides, although this work provides the theoretical guarantees to recover the true object positions up to quantifiable small shifts, there is no analysis whether it only exists in the unsupervised domain,.or can be adopted in the supervised domain.

**Questions:**

1. In the experiments, the datasets for evaluation is the CLEVR data, please explain why choose it, not other popular object detection datasets?
2. This work validate the theoretical results using lots of experimental results, however only few experiments are carried out for the comparison with SOTA.
3. In object detection, the popular model is about YOLO, and also the metric including accuracy, mAP, and IoU, etc are also the common in supervised object detection.

---

> ### Author Rebuttal · Authors · 2024-08-05
>
> > In the experiments, the datasets for evaluation is the CLEVR data, please explain why choose it, not other popular object detection datasets?
>
> We chose to base our dataset on CLEVR because it is a dataset commonly used in unsupervised learning, and because it allows us to generate images of the same object at different positions and with different sizes on a static background, which are the assumptions in our model. These assumptions do not necessarily hold in other datasets such as MS COCO or PASCAL VOC, which normally do not contain the same object twice, contain varying backgrounds, and do not have any bounds on the object sizes. In general, our method is more suitable for dynamic object detection from videos (where objects keep their identity but change position) rather than detecting single objects in images.
>
> > This work validate the theoretical results using lots of experimental results, however only few experiments are carried out for the comparison with SOTA.
>
> The aim of our paper is to present the first unsupervised object detection method with theoretical guarantees, and therefore our priority was to prove and validate our theoretical claims in detail. Our focus was less on the current SOTA methods, as, to the best of our knowledge, none of them possess any theoretical guarantees and thus are not directly comparable. We have compared with two current SOTA methods SAM and CutLER, but if the reviewer has any other particular method in mind, we are open to performing a comparison with that method.
>
> > there is no analysis whether it only exists in the unsupervised domain,.or can be adopted in the supervised domain
>
> For our method, deriving theoretical bounds was possible due to the exact equivariance property of the encoder and the decoder, and the restricted form of our latent space. In the supervised domain, many approaches are based on relatively complex transformer-based models, which do not possess any such guarantees. However, we speculate that to obtain guarantees for these methods, one might have to replace parts of their architectures with equivariant or invariant elements and introduce tight bottlenecks, in order to reduce the solution space sufficiently to enable the kind of theoretical analysis we have performed. We hope that our contribution will open up the possibility for such guarantees in future work.
>
> > In object detection, the popular model is about YOLO, and also the metric including accuracy, mAP, and IoU, etc are also the common in supervised object detection.
>
> Compared to YOLO, our method is fully unsupervised and possesses theoretical bounds on the detection errors, while theirs requires supervision and does not possess any such guarantees. While in our paper we prove bounds for the maximum position errors, these can be related to other metrics such as IoU by considering the overlap instead of the distance between the ground truth object and its detection (intuitively, the IoU bounds would be related to the square of the position bounds). We thank the reviewer for this suggestion and will add this to the paper.

---

> > ### Comment · Reviewer_pKP5 · 2024-08-12
> >
> > I would like to thank the authors for their responses. I will maintain the current score.

---

### Official Review · Reviewer_x9ub · 2024-07-12

**Soundness:** 2
**Presentation:** 2
**Contribution:** 2
**Rating:** 5
**Confidence:** 5

**Summary:**

The paper proposes a new idea for unsupervised object detection where an CNN based auto-encoder architecture is employed and the latent representation is trained to learn position of objects in images. They further provide theoretical analysis of the proposed idea under strong assumption about the input data and model characteristics. Results from on synthetic data experiments is also provided

**Strengths:**

The idea presented in the paper is interesting as it tries to solve the object detection problem in an unsupervised manner by modeling the latent space such that it explicitly learns object positions.

**Weaknesses:**

The paper lacks results and discussion on the experimental details on how the idea can be effectively implemented. This is particularly important to understand the merits of the proposed idea as it has strong assumption on model architecture and input data (e,g, size of objects). For example, it is not clear how the authors processes input data during training, how the min-batch sampling is done, what input-target pairs are?, what regulations are important to use if at all, how the over-fitting is prevented given the very simplified experimental setting.
Furthermore, it is not clear from the paper how the latent space can learn any semantic information to reconstruct the images as it modeled to learn the position of the objects.

**Questions:**

- Can the authors provide more clarification on the training procedure and the important aspects that are necessary for the model work?
- Given that the latent space is learning the position encoding for the object, how is it possible to learn to model semantics for reconstruction loss?
- how does the model performance change relative to diversity of object shape and appearance in a single image?
- why is it important to use positional encoding?
- how the reconstruction quality can be guaranteed, especially in a realistic setting?

**Limitations:**

- The work is limited by its assumptions on the characteristics of the input data and model architecture.
- The theoretical analysis is dependant on strong assumptions like "the objects are reconstructed at the same positions" which itself is not guaranteed.
- Furthermore, the evaluations do not provide insight into what challenges one should address to successfully train a model based on the proposed idea.
- Please see the Weaknesses for more details.

---

> ### Author Rebuttal · Authors · 2024-08-05
>
> > Can the authors provide more clarification on the training procedure and the important aspects that are necessary for the model work? For example, it is not clear how the authors processes input data during training, how the min-batch sampling is done, what input-target pairs are?, what regulations are important to use if at all
>
> We provide all training details for our experiments in Appendix C. To directly address the reviewer’s questions, in our synthetic experiments we train on images containing white squares of size 7x7 px at different positions on a background of size 80x80 px, with mini-batch size of 128 and learning rate $10^{-3}$, and the input-target pairs both being the same image. In our CLEVR experiments, we train on images containing spheres with different range of sizes (from 4 to 27 px) and at different positions, on a background of size 106x80 px, with mini-batch size of 150 and learning rates $10^{-2}$ and $10^{-3}$. In both cases we do not use any regularisation and train until convergence, discarding any result where an object has not been learned.
> > how the over-fitting is prevented given the very simplified experimental setting.
>
> Overfitting is prevented by the translation equivariance property of the architecture: an equivariant encoder and decoder, with a small receptive field, must reconstruct well at every possible sub-window of the image (as they are applied convolutionally). Equivariant models can thus learn from very few images compared to other architectures.
>
> > Given that the latent space is learning the position encoding for the object, how is it possible to learn to model semantics for reconstruction loss?
>
> Semantics are encoded in the ordering of the objects’ positions in the latent space: it is not an unordered list, but rather each position corresponds to a single object identity. Therefore, the encoder and decoder learn to associate each latent position to a different object, which allows the decoder to successfully reconstruct each object.
>
> > how does the model performance change relative to diversity of object shape and appearance in a single image?
>
> We found the model to be robust to lighting and perspective distortions of the objects in the CLEVR dataset (objects becoming larger/smaller when they are closer/further away). Although not included in the paper, the model was also successful in detecting different shapes (spheres, cubes, cylinders) at different orientations – please see figure 12 of the rebuttal PDF.
>
> > why is it important to use positional encoding?
>
> We use positional encodings as one input to the decoder to reconstruct a static background. Because the decoder is translation equivariant, the positional encodings provide it with the positional information necessary to successfully reconstruct different parts of the background. For dynamic backgrounds, an unrelated frame from the same video can be passed instead of the positional encoding (as mentioned in section 3).
>
> > The work is limited by its assumptions on the characteristics of the input data and model architecture.
>
> The assumptions on the input data and model architecture are necessary for our proof of the theoretical bounds of the method – it is impossible to derive any guarantees with no assumptions. We believe these assumptions are not very restrictive and apply to many common use cases for object detection such as traffic monitoring, surveillance, etc, where there are moving objects and a static background (we also mention an extension for dynamic backgrounds in section 3).
>
>
> > how the reconstruction quality can be guaranteed, especially in a realistic setting?
> > The theoretical analysis is dependant on strong assumptions like "the objects are reconstructed at the same positions" which itself is not guaranteed.
>
> This is guaranteed by the Universal Approximation Theorem: a sufficiently-large neural network can approximate any function. If the reconstruction error is high, we simply need a larger network, until the error is low enough.
>
> > Furthermore, the evaluations do not provide insight into what challenges one should address to successfully train a model based on the proposed idea.
>
> In practice, successful training requires videos with a static camera, objects that fit the receptive field of the autoencoder, that stay in-frame for the duration of a video, and no large perspective changes during motion. We provide further training details in Appendix C.

---

> > ### Comment · Reviewer_x9ub · 2024-08-13
> >
> > Thank you for answering the questions; I have raised my rating given the clarifications, However I strongly recommend the authors to update the paper with the clarifications they have provided, in particular on overfitting, how the ordering of the objects’ positions are key for the latent space, how sensitive is the proposed solution to object shape and appearance, and why including positional encoding is important.

---

### Official Review · Reviewer_YrtR · 2024-07-15

**Soundness:** 3
**Presentation:** 2
**Contribution:** 3
**Rating:** 6
**Confidence:** 5

**Summary:**

This paper presents the first unsupervised object detection approach that is theoretically shown to recover the true object positions up to quantifiable small deviations that are related to the encoder and decoder receptive field sizes, the object sizes, and the widths of the Gaussians used in the rendering process. The authors conduct a thorough analysis of how the error depends on each of these variables and conduct synthetic experiments that validate our theoretical predictions up to a precision of individual pixels.
On a high level, their architecture is based on an autoencoder that is fully equivariant to translations, which they achieve by making the encoder consist of a CNN followed by a soft argmax function to extract object positions, and making the decoder consist of a Gaussian rendering function followed by another CNN to reconstruct an image from the object positions.
The authors also conducted synthetic experiments, CLEVR-based experiments, and real video experiments that validated their theoretical findings up to a precision of individual pixels.

**Strengths:**

I do like the analysis of the current state-of-the-art detection models SAM and CutLER and it is interesting to find that in some cases their errors are much higher than the bound derived by this method.

This paper is well-written and easy to follow.

**Weaknesses:**

1. It is interesting to learn that SAM and CutLER's errors are sometimes much higher than the bound derived by the proposed method. I would be interested to hear from the authors if they have any insights on how this finding could be used to improve these methods, especially CutLER, which is also an unsupervised object detection and instance segmentation model.

2. The majority of the experiments in this paper are conducted on synthetic datasets, and it is questionable whether the findings can be generalized to real images and videos. Could the authors provide some experiments on real images or videos?

3. Continuing on the previous point, most objects in the synthetic datasets are rigid and have very consistent shapes. However, the challenges in object detection are often in detecting the non-rigid objects or partially occluded objects. I am curious to see if the proposed method can be used to handle these cases.

**Questions:**

Please check the weakness section

**Limitations:**

Yes

---

> ### Author Rebuttal · Authors · 2024-08-05
>
> > It is interesting to learn that SAM and CutLER's errors are sometimes much higher than the bound derived by the proposed method.
>
> We note that all the error plots in the paper contain the maximum position errors, as opposed to average position errors (as described in Appendix C). So, while SAM and CutLER do well in the average case, they are much worse (unbounded) than our method in the worst case. We do this to show that even in the worst case our method is able to contain the errors within our bounds unlike other methods.
>
> > I would be interested to hear from the authors if they have any insights on how this finding could be used to improve these methods, especially CutLER, which is also an unsupervised object detection and instance segmentation model.
>
> For our method, deriving theoretical bounds was possible due to the exact equivariance property of the encoder and the decoder, and the restricted form of our latent space. However, because both SAM and CutLER are relatively complex transformer-based models, it is difficult to derive theoretical bounds for these methods. However, we speculate that to obtain guarantees for these methods, one might have to replace parts of their architectures with equivariant or invariant elements and introduce tight bottlenecks, in order to reduce the solution space sufficiently to enable the kind of theoretical analysis we have performed. We hope that our contribution will open up the possibility for such guarantees in future work.
>
> > The majority of the experiments in this paper are conducted on synthetic datasets, and it is questionable whether the findings can be generalized to real images and videos. Could the authors provide some experiments on real images or videos?
>
> We perform experiments on real videos in Appendix D of the paper, which are based on real YouTube videos of overhead road traffic and a game of mini pool. In these experiments we show that the objects are learned with high precision and the latent space can be intervened on to generate realistic videos with the objects at previously unseen positions. We will be happy to add more if requested.
>
> > Continuing on the previous point, most objects in the synthetic datasets are rigid and have very consistent shapes. However, the challenges in object detection are often in detecting the non-rigid objects or partially occluded objects. I am curious to see if the proposed method can be used to handle these cases.
>
> As this is the first paper to propose any theoretical guarantees for object detection, we decided to focus on producing a detailed analysis of the common case of rigid, non-occluded objects. However, while not discussed in the paper, we observe in our experimental data that the method is relatively robust to partial occlusions (e.g. one sphere covering part of another sphere in CLEVR). Regarding non-rigidity, we believe that as long as the deformed object fits within the receptive field, the method should also be able to detect those objects as long as they are sufficiently similar to the original object. For more complex cases of occlusions and non-rigidity, one might have to amend the method to deal with these cases explicitly (e.g. by modeling the order in which objects are rendered, and modeling the explicit geometry of non-rigid objects). We leave these for future work, to maintain the conciseness and clarity of the current proofs.

---

> > ### Comment · Reviewer_YrtR · 2024-08-12
> >
> > I would like to thank the authors for their responses. Most of my questions have been addressed. Therefore, I will maintain my current rating as weak accept.

---

### Author Rebuttal · Authors · 2024-08-06

In response to reviewer’s ghv9 question, we have performed CLEVR experiments showing the position error as a function of the Gaussian standard deviation (see fig. 11 in the rebuttal PDF). As all our data points (red) lie within our theoretical bounds (blue), this successfully validates our theory.

Additionally, in response to reviewers’ x9ub and ghv9 questions, we have performed CLEVR experiments using a dataset containing 3 different shapes (a sphere, a cube, and a cylinder), showing that our method works for objects of any shape, not just spheres (see fig. 12 in the rebuttal PDF).

We hope that these experiments address the reviewers’ concerns.

---

### Decision · Program_Chairs · 2024-09-25

**Decision:**

Accept (poster)

**Comment:**

This paper is about unsupervised object detection, the authors propose a way to build an unsupervised object detection with theoretical guarantees, meaning known bounds on position errors, via an autoencoder architecture with latent variables.

The paper contributes to the start of the art with a unsupervised object localization architecture that can recover true object positions to a known error bound, and theoretical analysis to demonstrate error bounds, and how these bounds depend on encoder/decoder receptive field sizes, object size, and gaussian variance during rendering. Experiments are made on synthetic data generated by the authors and on CLEVR dataset, and show the validity of the bounds.

All reviewers agree that the paper is very well written and mostly easy to understand/follow, there is a good evaluation and a comparison to previous methods (SAM and cutLER), and the approach is in the right direction to solve the unsupevised object detection problem, with an innovative and novel approach.

One minor weakness is that all experiments are made on synthetic data, and its not clear how this method would perform in real data.

Some minor improvements are suggested for this paper, the proposed method actually performs object localization, as the whole theoretical guarantee is about object localization, and object detection also includes object classification which is not considered in this approach, clarify the training process, and the limitations and assumptions of the proposed approach. Also please consider the many small details mentioned by the reviewers.

Overall this paper is above the acceptance threshold, and should be accepted.